# Chemical Characterization and In Vivo Toxicological Safety Evaluation of Emu Oil

**DOI:** 10.3390/nu14112238

**Published:** 2022-05-27

**Authors:** Meijuan Lan, Lin Li, Shengkai Luo, Juncheng Chen, Xiaofeng Yi, Xia Zhang, Bing Li, Zhiyi Chen

**Affiliations:** 1Sericultural & Agri-Food Research Institute, Guangdong Academy of Agricultural Sciences Key Laboratory of Functional Foods, Ministry of Agriculture and Rural Affairs, Guangdong Key Laboratory of Agricultural Products Processing, Guangzhou 510610, China; 18959950300@163.com; 2College of Food Science and Engineering, Guangdong Province Key Laboratory for Green Processing of Natural Products and Product Safety, Engineering Research Center of Starch and Plant Protein Deep Processing, Ministry of Education, South China University of Technology, Guangzhou 510640, China; lilin@dgut.edu.cn (L.L.); m13760723613@163.com (S.L.); chenjuncheng2009@163.com (J.C.); cexzhang@scut.edu.cn (X.Z.); 3School of Chemical Engineering and Energy Technology, Dongguan University of Technology, Dongguan 523808, China; 4Department of Chemical and Biochemical Engineering, College of Chemistry and Chemical Engineering, Key Laboratory for Chemical Biology of Fujian Province, Xia-men University, Xiamen 361000, China; 20620180155122@stu.xmu.edu.cn

**Keywords:** emu oil, fatty acid, antioxidant activities, toxicological safety

## Abstract

In this study, the physicochemical properties, fatty acid composition, antioxidant activities, and in vitro as well as in vivo toxicological safety of emu oil were investigated. Emu oil was shown to have a low acid and peroxide value, low amounts of carotenoid and phenolic compounds, and high doses of oleic acid and linoleic acid. Furthermore, in a bacterial reverse mutation assay, emu oil demonstrated no change in the amount of revertant colonies for all strains. In a chromosomal assay, no aberrations occurred in any of the emu oil treatment groups (1.25, 2.5, and 5 μg/mL). In the bone marrow micronucleus test, emu oil up to 20 mL/kg showed no significant increase in the incidence of micronucleated polychromatic erythrocytes. Moreover, emu oil up to 19.3 mg/kg body weight did not affect body weight in an acute oral toxicity study. These results are crucial for the adoption of emu oil as an alternative source of edible oil.

## 1. Introduction

Animal fats possess desirable technological properties and are widely used in certain traditional bakery foods to endow products with a uniquely appealing flavor, which is attributable to the triacylglycerides (TAG) and fatty acid (FA) compositions in solid fat [1]. For example, tallow is popular in bakery products; it endows the products with a special and fascinating flavor. There are a variety of animal fats that can roughly be divided into liquid fats (fish oil) and solid fats (lard and tallow) [2]. The high cholesterol content of lard results in a potential risk for cardiovascular diseases in consumers. Moreover, the lard cannot be used in halal or kosher products. Therefore, it is necessary to develop new fats to expand the application of animal fats in food processing.

The emu (*Dromaius novaehollandiae*) originates in Australia and has the highest proportion of adipose tissue in ratites [3]. The leading countries that rear emu are the United States and China. Most ratites are consumed and traded in markets across the globe for the production of meat and by-products; oil is a major by-product of the emu process industry [4]. Emus accumulate oil mainly under the skin and in the abdominal cavity [5]. Emu oil is a complex mixture of lipids, of which the main component is triacylglycerol (TAG). Therefore, FAs in the human diet can be gained from emu oil. Moreover, the presence of minor components, such as chlorophylls [6], phenolics [7], or carotenoids [8], endows emu oil with excellent anti-inflammatory effects via a significant reduction in the levels of proinflammatory cytokines, including TNF-α and IL-1α [9]. Emu oil has been shown to prevent and treat gastrointestinal inflammation and bone problems after oral administration in an animal model [10,11]. In addition, it can be used in cosmetics and oil-based pharmaceuticals for external use, primarily on account of its anti-inflammatory and moisturizing effects [10]. Although emu oil is ingested with the consumption of emu meat in a dietary capacity, as a new animal oil source, toxicological studies on emu oil in food are still lacking.

Here, the physicochemical properties, fatty acids, bioactive components (phenols and β-carotene), and antioxidant activity of emu oil were determined. We preferentially compared the chemical properties of emu oil and common animal cooking oils from daily life, such as lard and tallow, to promote the application of emu oil in daily life. Moreover, in order to further promote the safe use of emu oil, we also investigated the mutagenicity and toxicity of emu oil using micronucleus, bacterial reverse mutation, and chromosomal aberration tests in vivo for the first time. Finally, we attempted to elucidate its security and reliability for food processing.

## 2. Materials and Methods

### 2.1. Materials and Reagents

Emu fatty tissue was obtained from Guangdong Xinji Emu Industry Co., Ltd. (Jiangmen, China). The β-carotene, gallic acid, Folin–Ciocalteau reagent, 2,2-diphenyl-1-picrylhydrazyl (DPPH), 2,2′-azino-bis (3-ethylbenzothiazoline-6-sulphonic acid) (ABTS), cytochalasin B, penicillin GNa, ethyl methanesulfonate, 2-aminofluorene, trypsin-EDTA, 4-nitro-1,2-phenylenediamine monohydrochloride, 1,8-dihydroxyanthraquinone, mitomycin C, 4-nitroquinoline-N-oxide, linoleic acid, tertbutyl hydroquinone (TBHQ), cyclophosphamide monohydrate, and Giemsa were obtained from Sigma-Aldrich (Sigma Aldrich Trading Co., Ltd., Steinheim, Germany). Corn oil was obtained from Yihai (Guangzhou) Grain and Oil Industry Co., Ltd. (Guangzhou, China). The malondialdehyde (MDA) assay kit was obtained from Beijing Solarbio Science & Technology Co., Ltd. (Beijing Solarbio Science & Technology Co., Ltd., Beijing, China). All the reagents used were of analytical grade.

### 2.2. Extraction of Emu Oil by Super Critical Fluid Extraction

The extraction of emu oil by super critical fluid extraction was carried out according to a previously published protocol with minor modifications [12]. Briefly, 50 g of emu fatty tissue was placed in the 100 mL extraction vessel of the OV-SCF-10000 Series supercritical fluid extraction apparatus (Taichao Extraction Cleaning Machine Co., Ltd., Penghua, China). The temperature and pressure were maintained at 45 °C and 35 MPa, respectively. The CO_2_ flow rate was 4 L·min^−1^ and the total extraction time was around 60 min. Finally, the extracted emu oil was collected and stored at −18 °C until further analysis.

### 2.3. Fatty Acid Composition of Emu Oil

The determination of fatty acid composition was based on the procedure referenced in a previous study [13]. Fatty acid methyl esters were determined using an Agilent 7890 gas chromatograph (Agilent Technology, Santa Clara, CA, USA) and a flame ionization detector equipped with a DB-5 MS column (30 m × 0.25 mm × 0.25 μm). The carrier gas was He with a flow rate of 1 mL/min. The oven temperature was maintained at 150 °C for 4 min and subsequently heated to 280 °C, increasing by 4 °C/min. Finally, the fatty acid methyl esters were detected by comparing the retention times with a commercial standard obtained from Sigma-Aldrich (Sigma Aldrich Trading Co., Ltd., Steinheim, Germany).

### 2.4. Physicochemical Properties of Emu Oil

The acid value (Cd 3d-63), peroxide value (Cd 8-53), iodine value (Cd 1-25), and free fatty acid contents (Ca 5a-40) were measured according to the AOCS methods [14].

The malondialdehyde (MDA) level was analyzed by using a malondialdehyde (MDA) assay kit (Beijing Solarbio Science & Technology Co., Ltd., Beijing, China).

The total unsaponifiable substance was measured in accordance with the previously described protocol [15]. Briefly, 5 g of emu oil was mixed with 2.2 N potassium hydroxide solution in an ethanol–water mixture (8:2, *v*/*v*). Then, saponification was carried out by boiling and stirring the sample for 1 h. After cooling, the sample was extracted using ethyl ether. Finally, the ether extract was washed, dried, and weighed.

A WYA-2S Abbe refractometer (Shanghai Yidian physical optical instrument Co., Ltd., Shanghai, China) was used to determine the refractive index of emu oil in accordance with the method described in [12].

The carotenoid content of the emu oil was detected using a previously reported method with some modifications [16]. 3 g of emu oil was dissolved in 10 mL of hexane, and a Cary 50 Conc UV-Vis spectrophotometer (Varian Australia Pty. Ltd., Melbourne, Australia) was used to detect the absorbance at 468 nm. The carotenoid content was calculated as follows:$$\mathrm{Carotenoid}\;\left(\mathrm{mg}/\mathrm{kg}\right)=\frac{A_{470}\times 10^{6}}{2000\times 100\times d}$$
where *A*_470_ refers to the absorbance of the emu oil samples at 470 nm; the molecular absorption of lutein is represented by the value 2000; and *d* is the thickness of the cuvette.

The determination of total phenolic content was carried out according to a previously described method [17]. Briefly, 2 g of emu oil was mixed with 5 mL of hexane and extracted with 5 mL of methanol/deionized water (60:40, *v*/*v*). Then, the solution was vortexed for 5 min. After that, 0.5 mL of Folin–Ciocalteu reagent and 4.8 mL of water were mixed. The reaction was carried out in the dark for 2 h; then, an Infinite M200 microplate reader (Tecan Trading Co., Ltd., Männedorf, Switzerland) was used to measure the absorbance at 725 nm. The total phenolic content was expressed in mg gallic acid/kg oil.

### 2.5. Antioxidant Activities

#### 2.5.1. DPPH Scavenging Activity

The DPPH radical scavenging activity of emu oil was evaluated in accordance with a protocol from a previous study, with some alterations [18]. Briefly, 100 μL of oil samples (25, 50, 100, 200, and 400 mg/mL) was reacted with 100 μL of 0.10 mM DPPH solution in the dark for 30 min at room temperature. An Infinite M200 microplate reader (Tecan Trading Co., Ltd., Männedorf, Switzerland) was used to measure the absorbance of the mixture at 517 nm of mixture sample. The DPPH radical scavenging rate (%) was calculated by:$$\mathrm{DPPH}\;\mathrm{radical}\;\mathrm{scavenging}\;\mathrm{rate}\;\left(\%\right)=\frac{A_{blank}\times A_{sample}}{A_{blank}}\times 100$$
where *A**_blank_* refers to the absorbance of the mixture without the oil sample and *A**_sample_* represents the absorbance of the mixture containing the oil sample.

#### 2.5.2. ABTS Scavenging Activity

The ABTS scavenging activity of emu oil was assessed using a previously reported protocol with some modifications [19]. A mixture of 7 mM ABTS solution and 2.45 mM potassium persulfate was applied to induce the generation of the ABTS radical cation. The ABTS scavenging activity was measured by mixing 100 μL of oil samples (200, 400, 600, 800, and 1000 mg/mL) with 100 μL of diluted ABTS solution and reading the absorbance (734 nm) after 6 min using an Infinite M200 microplate reader (Tecan Trading Co., Ltd., Männedorf, Switzerland). The results were expressed as an ABTS radical scavenging rate (%), which was then calculated as follows:$$\mathrm{ABTS}\;\mathrm{radical}\;\mathrm{scavenging}\;\mathrm{rate}\;\left(\%\right)=\frac{A_{control}\times A_{sample}}{A_{control}}\times 100$$
where *A**_control_* is the absorbance of the ABTS solution without an oil sample and *A**_sample_* is the absorbance of the ABTS solution with emu oil.

### 2.6. In Vitro and In Vivo Toxicological Assessment of Emu Oil

#### 2.6.1. Bacterial Reverse Mutation Study (Ames Test)

The plate incorporation method was used to perform an Ames test [20]. The Salmonella typhimurium strains TA97a, TA 98, TA100, TA102, and TA1535 were incubated with emu oil in DMSO at doses of 50 μg/plate, 158.1 μg/plate, 500 μg/plate, 1581 μg/plate, and 5000 μg/plate, respectively, in the presence or absence of metabolic activation. In the presence of metabolic activation, 2-aminofluorene (5 μg/plate) was used as a positive control for the TA97a, TA98, TA100, and TA1535 groups, while the positive control for the TA102 group was treated with 1,8-dihydroxyanthraquinone (50 μg/plate). In the absence of metabolic activation, 4-nitro-1,2-phenylenediamine monohydrochloride (40 μg/plate for the TA97a and TA100 groups), 2-aminofluorene (5 μg/plate; for the TA98 group), Mitomycin C (1 μg/plate for the TA102 group), and 4-nitroquinoline-N-oxide (1 μg/plate for the TA1535 group) were used as the positive control treatments. The spontaneous revertant colonies of all strains were counted, while DMSO solution was used as a negative control.

#### 2.6.2. Extracorporeal Mammalian Chromosome Aberration Test

To evaluate the potential genotoxicity of emu oil, we carried out the mammalian chromosome aberration tests based on the procedure described in a previous report [21] with slight modifications. Chinese hamster lung (CHL) cells were provided by the National Collection of Authenticated Cell Culture (Chinese Academy of Sciences, Shanghai, China). The CHL medium was composed of a minimum essential medium (MEM) containing 10% fetal bovine serum. The incubation of the CHL cells was performed on plastic plates at a density of 1 × 105 cells/mL for 24 h. The Emu oils were diluted with MEM at doses of 1.25, 2.5, and 5 μL/mL, with or without the metabolic activation. The incubation of all cells was performed for 2 h with or without the addition of the S9 mix. MEM, mitomycin C, and cyclophosphamide monohydrate were treated as the negative control, positive control without metabolic activation, and positive control with metabolic activation, respectively. Then, phosphate-buffer saline (PBS) was used to wash the cells 3 times, and the cells were allowed to incubate in the MEM for 22 h. In addition, 1 μg/mL of the metaphase arresting agent colchicine was added 4 h prior to cells collection. Then, 0.25% trypsin was added to the collected cells and centrifugation of cells was carried out at 1000 rpm, followed by the addition of 75 mM potassium chloride solution to cell collection. The fixation and washing of cells were performed using a methanol and acetic acid (3:1, *v*:*v*) solution. The slides were stained with 3% Giemsa (Zhuhai Besuo Biotechnology Co., Ltd., Zhuhai, China) for examinations using a BX41 fluorescence microscope (Olympus Corporation, Tokyo, Japan).

#### 2.6.3. Animals

NIH mice (10–12 weeks old, 20.6–28.0 g) were provided by the Guangdong Medical Laboratory Animal Center (Guangzhou, China) and raised in controlled conditions at a temperature of 23 ± 3 °C under a 12 h light–dark cycle. The necessary approvals were obtained from the Animal Ethics Committee of Guangdong Medical Laboratory Animal Center (Approval No. A202003-6, approval date: 30 March 2020).

#### 2.6.4. Micronucleus Test (MN)

A micronucleus test was carried out using Hwang’s method with some modifications [22]. The in vivo micronucleus test was performed to evaluate the potential for emu oil to induce an increase in micronucleated polychromatic erythrocytes in NIH mice (Guangdong Medical Laboratory Animal Center, Guangzhou, China). Both genders were distributed into 5 groups (n = 10/group). The vehicle control and positive control groups were treated with corn oil and cyclophosphamide (4 mg/mL), respectively. The emu oil was diluted with corn oil to achieve doses of 25%, 50%, and 100%. In addition, the vehicle control and emu oil group were given an orally administered volume of 20 μL/g body weight of corn oil and emu oil, respectively. Meanwhile, the mice in the positive control group were intraperitoneally injected with cyclophosphamide at 10 μL/g body weight. The vehicle, positive control, and emu oil were administered twice at approximately 24 h intervals. Bone marrow smears were obtained from all animals 6 h after the second injection. Finally, the smears were stained with Giemsa stock solution and then washed in distilled water for 4 min.

The stained slides were observed using a microscope with an oil immersion lens at 100× X magnification. The coded slides were analyzed under a microscope, and the percentage of polychromatic erythrocytes (PCEs) among 200 normochromatic erythrocytes (NCE) was identified. Based on the observation of 2000 PCEs in each slide, the quantity of micronucleated PCEs was recorded for the cytotoxic assessment of each sample.

#### 2.6.5. Acute Oral Toxicity Study

The acute toxicity study of emu oil was performed in conscious mice with slight modifications to the method described in [23]. The mice were divided into 2 groups (n = 10/group) according to sex. Then, they were fasted for 6 h, weighed, and treated by orogastric gavage with emu oil at a dose of 19.3 mg/kg body weight. The observation of animals was carried out by recording any clinical signs or mortality at 30 and 240 min, and once daily for 14 days. On the 7th and 14th days, the animals undergoing the emu oil treatment were weighed again to determine their weight variation during the experiment. Finally, a mixture of ketamine and xylazine was applied to anesthetize the animals via the intraperitoneal route.

### 2.7. Statistical Analysis

The results were expressed as the mean ± SD in triplicate. Duncan’s multi-range tests were used to statistically analyze the differences among groups with SPSS software (SPSS Inc., Chicago, IL, USA). The statistical differences in activity were expressed via the use of manuscript letters for different treatments. In addition, *p* < 0.05 and *p* < 0.01 were considered to be statistically significant.

## 3. Results

### 3.1. Physicochemical Properties of Emu Oil

The determined physicochemical properties of emu oil are shown in Table 1. The acid value and the free fatty acids (FFAs) content of emu oil were 1.24 ± 0.49 mg KOH/g and 0.8 ± 0.07%, respectively. Additionally, the FFA content of emu oil was lower than that of the FFA value of lard and tallow, as shown in Table 1. Moreover, the MDA value of emu oil was 0.02 ± 0.01 mg/100 g oil, which was a lower percentage than that of lard [24]. At the same time, the peroxide value was 0.42 ± 0.41 g/100 g oil, which was also lower than that of lard. The refractive index and iodine value of emu oil were 1.46 ± 0.01 and 72.67 ± 2.08 g/100 g oil, respectively. However, the iodine value of emu oil was higher than that of lard and tallow.

In addition, the emu oil contained 0.54 ± 0.13% unsaponified matter. As shown in Table 1, the carotenoid content in emu oil was 5.92 ± 0.62 mg/kg oil. The total polyphenol content of emu oil was 6.64 ± 0.37 mg/kg. Furthermore, carotenoid and phenolic content was not found in the lard or tallow.

### 3.2. Fatty Acid Profile of Emu Oil

The fatty acid composition of emu oil is shown in Table 2. We mainly identified eight kinds of fatty acids. The saturated fatty acid content of emu oil was 34.78%, which is lower than that of lard or tallow (Table 2). In addition, the unsaturated fatty acids prevailed in emu oil (64.28 ± 1.04%), with oleic acid (45.76 ± 0.53%) being the highest content of unsaturated fatty acids. Linoleic acid was the second most abundant unsaturated fatty acid in emu oil (14.00 ± 0.21%). Furthermore, the contents of oleic acid and linoleic acid in emu oil were richer than those of lard and tallow.

### 3.3. Antioxidant Activity

The free radical scavenging abilities of antioxidants can be assessed by DPPH, which is an organic compound composed of stable free radical molecules [16]. Here, TBHQ was used as a positive control to assess the antioxidant activity of emu oil. As shown in Figure 1a, the DPPH scavenging rate of emu oil increased with the increasing oil concentration, and the highest DPPH scavenging activity rate of 74.21 ± 3.12% was found at a concentration of 400 mg/mL. However, the DPPH scavenging rate of emu oil was lower than that of TBHQ solution at the same concentration.

The antioxidant activity of emu oil was further verified using an ABTS assay. In order to evaluate the antioxidant activity of emu oil, the control was treated with TBHQ. As evident in Figure 1b, similarly, the ABTS scavenging rate of emu oil had a strong positive correlation with a concentration over the range of 200–1000 mg/mL, and finally reached a maximum of 72.50 ± 1.42%.

### 3.4. In Vitro and In Vivo Toxicological Evaluation of Emu Oil

#### 3.4.1. Bacterial Reverse

The in vitro toxicological safety of emu oil was evaluated using a bacterial reverse mutation assay with five bacterial strains (TA97a, TA98, TA100, TA102, and TA1535; Table 3). There was no significant difference in the number of revertant colonies between the vehicle control groups and the groups at any dosage level of emu oil (*p* > 0.05). However, the numbers of revertant colonies in the positive control were significantly higher than those in the negative control (*p* < 0.05). The positive controls with or without S9 mix showed a three-fold increase in revertant colonies over the vehicle control. An additional confirmatory test was used to verify the previous results with emu oil at a concentration of up to 5000 μg/plate (Table 3). In all bacterial strains, there was no substantial increase change in the revertant colonies between the vehicle control group and emu oil groups in quantitative terms (*p* > 0.05).

#### 3.4.2. In Vitro Mammalian Chromosome Aberration Assay

As shown in Figure 2, different types of chromosomal aberrations were observed in the positive control group (mitomycin C and cyclophosphamide), including chromatid exchanges, rings, translocations, and chromatid breaks. The obvious positive reaction indicated that the system was reliable under the test conditions.

For the samples without S9 mixed, no significant difference was noted in the emu oil groups or MEM group (*p* > 0.05). Moreover, the chromosome aberration rates for the different doses of emu oil (5.00, 2.50, and 1.25 μL/mL) were lower than those of the mitomycin C group (positive control). The chromosome aberration rate of the mitomycin C group (0.4 μg/mL) was 10.0%, and the chromosome aberration rate was higher than that of both the MEM group and the emu oil group (*p* < 0.01) (Table 4).

For the samples with S9, the chromosome aberration rates for the different doses of emu oil were lower than that of cyclophosphamide, and the difference between the emu oil and MEM group remained nuanced (*p* > 0.05). As for the positive control, the chromosome aberration rate of cyclophosphamide was also 10.0%. The positive control induced a statistical increase in the number of chromosome aberrations compared with the MEM group (*p* < 0.01).

#### 3.4.3. Micronucleus Test

The effects of emu oil on the frequency of micronuclei in mouse bone marrow are presented in the Table 5. The polychromatic erythrocyte counts for the emu oil groups were not statistically different from those of the corn oil groups (*p* > 0.05). Moreover, no obvious differences in the induction of micronuclei were found between the groups supplemented with emu oil and corn oil (*p* > 0.05). Our evaluation of the incidence of MPCE indicated a pronounced cytotoxic effect of cyclophosphamide on the bone marrow erythroid compartment at a dose of 40 mg/kg body weight (*p* < 0.01). However, emu oil did not inhibit bone marrow cell proliferation.

#### 3.4.4. Acute Oral Toxicity

As shown in Figure 3 and Table 6, with 19.3 mg/kg emu oil at each feeding time (feeding dose to body weight), no clinical signs or death were observed among the mice tested. Moreover, no abnormality was denoted in the mice during the 14-day study period. Additionally, body weight increased in both the male and female groups (Figure 2).

## 4. Discussion

A chemical characterization and in vivo toxicological safety evaluation of emu oil was carried out in this study. The unesterified fatty acid levels of oil samples were assessed by the acid number and the contents of free fatty acids, thus defining their quality [31]. Here, emu oil possessed a low acid value and peroxide value. The lower acid value indicated that emu oil might have a long shelf life without deterioration [32]. At the same time, the FFA content of emu oil was lower than that of lard and tallow, which also indicated that the emu oil possessed a longer storage time than lard and tallow. In addition, the obtained MDA content of emu oil showed similar results. MDA was induced by the degradation of polyunsaturated fatty acids during lipid oxidation [33]. The peroxide value of emu oil was similar to the results reported by Gbogouri [34]. The peroxide value was used to measure the oxidative deterioration of oil, making it an important parameter in the characterization of the oil quality [35].

The refractive index and iodine values of emu oil were similar to those reported in a previous study on tiger nut oil [12]. The refractive index and iodine value were related to the fatty acid composition of oils. Here, the high content of unsaturated fatty acids caused a large refractive index and a high iodine value [36]. However, the iodine value of emu oil was higher than that of lard and tallow, as shown in Table 1, which was due to a lower percentage of saturated fatty acids in emu oil shown in Table 2.

Emu oil had functional properties due to its high amount of unsaturated fatty acids (mainly oleic acid) and phenolic compounds. Fatty acids not only influence the physicochemical characteristics of the emu oil, but also determine its biological effects after consumption [37]. It has been shown that oleic acid can be used to prevent breast cancer and rheumatoid arthritis. Meanwhile, the prothrombotic state of the postprandial phase may also be attenuated by oleic acid [38]. In addition, linoleic acid, which can decrease the risk of cardiovascular diseases, was shown to be the second most abundant unsaturated fatty acid in emu oil, which can decrease [39]. These results indicated that after consumption, emu oil was possibly more beneficial to the human body. In respect to chemical composition, emu oil contains more unsaturated fatty acids than lard and tallow. Moreover, it has biological components, such as polyphenols, carotenoids, etc. [40]. It was clear that emu oil could potentially be utilized as a resource for edible animal oil. The polyphenols may be responsible for the antioxidant activities observed in the present study. Polyphenols are a class of compounds with various biological activities, including antibacterial activity, antioxidant, and anti-inflammatory activities, among others [41,42]. Additionally, the unsaturated fatty acids in emu oil would scarcely react with free radicals [43].

Although emu oil is ingested with the dietary consumption of emu meat, as a new animal oil source, toxicological studies on emu oil in food are still lacking. It is necessary to prove the safety of emu oil before it can be used for the enrichment of dietary oils or other foods.

The bacterial reverse test is used to detect gene mutations caused by DNA damage. In our bacterial reverse mutation assay, emu oil (50, 158.1, 500, 1581, and 5000 μg/plate) did not increase the number of revertant colonies in any experimental strain with or without S9 mix. Similarly, the amount of revertant colonies in the bacterial strains treated with bayberry kernel oil at a dose level of 5000 μg/plate also did not exceed the mutagenic level indicative of mutagenicity [44]. Based on these results, we concluded that any test dose of emu oil showed non-mutagenic activity in the bacterial reverse assay.

An in vitro chromosome aberration assay was used to investigate the chromosome breakage potential of emu oil [45]. In a chromosomal assay, no statistical increase was found in either kind of aberration at a certain dose of emu oil in groups (1.25, 2.5, and 5 μg/mL) with or without S9 metabolic activation. Similar results were reported by Matulka et al. [46], who showed that no indistinctive increase in the aberrant cells’ incidence was found in samples supplemented with odd-chain fatty algal oil at a dose level up to 2000 mg/kg body weight for 24 or 48 h. These results demonstrated that emu oil could barely induce aberrant cells and cell toxicity both with and without S9 mix.

In an in vivo micronucleus test, emu oil up to 20 mL/kg body weight caused no substantial increase in the frequency of micronucleated polychromatic erythrocytes, or the rate of immature erythrocytes to total erythrocytes. Similarly, in another study, mouse bone marrow cells treated with neem oil for also showed no significant effects on genotoxicity compared to the control [22]. Therefore, our results indicated that emu oil had no cytotoxic effects.

Moreover, emu oil up to 19.3 mg/kg body weight did not affect body weight in the acute oral toxicity study. Previous reports were investigated by Rodríguez-Lara [47] who reported an increase in body weight between the mice of the vehicle control group and the mice treated with olive oil extract (300 mg/kg) for the 14 days.

In summary, the chemical characterization of emu oil was systematically investigated in the present study. Compared with lard and tallow, emu oil possessed a longer storage time and more bioactive unsaturated fatty acids. This indicated that emu oil not only had a higher potential to be applied in everyday life but was also more beneficial to the human body. Furthermore, the in vitro and in vivo toxicological analysis revealed that emu oil showed no mutagenesis, cytotoxicity, or acute oral toxicity. This further suggests that emu oil can be safely applied in the food industry. Therefore, the results of the present study are decisively in favor of the adoption of emu oil as an alternative source of edible oil and the further expansive application of emu oil in food processing.

## Figures and Tables

**Figure 1 nutrients-14-02238-f001:**
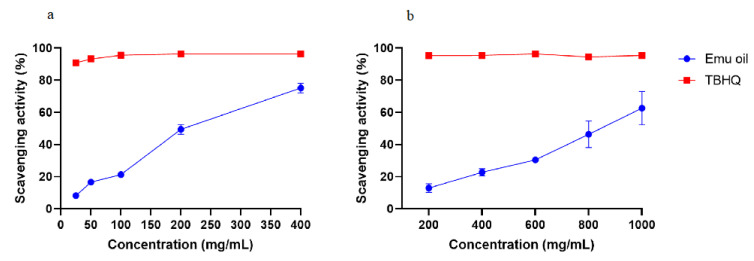
Radical scavenging activities of emu oil. (**a**) DPPH free radical scavenging activity assay, and (**b**) ABTS radical scavenging activity assay.

**Figure 2 nutrients-14-02238-f002:**
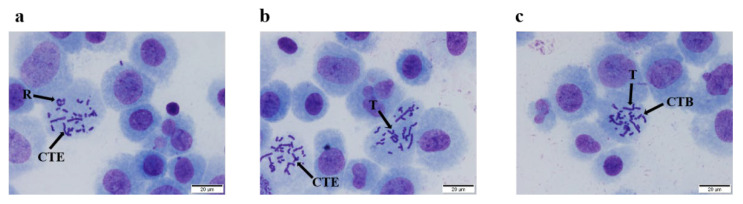
Types of chromosomal aberration for cell under microscope. (**a**). chromatid exchange (CTE) and ring (R); (**b**). chromatid exchange (CTE) and translocation (T); (**c**). chromatid break (CTB) and translocation (T).

**Figure 3 nutrients-14-02238-f003:**
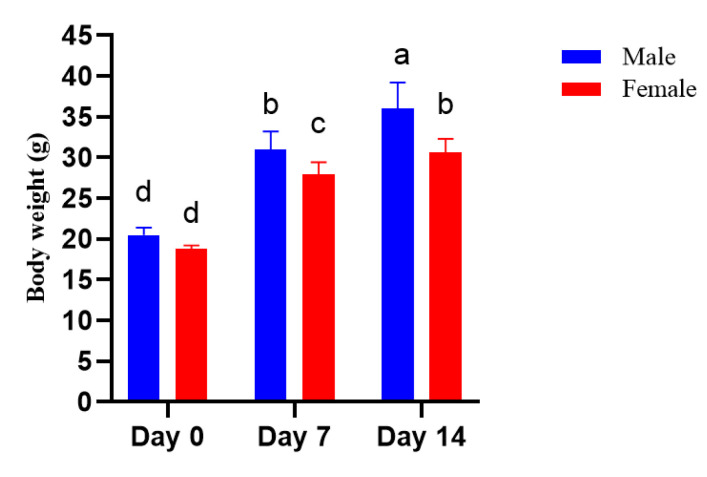
Body weight of mice supplemented with emu oil in the acute toxicity test. Letters on the bar within the same form of emu oil indicate significant differences (*p* < 0.05).

**Table 1 nutrients-14-02238-t001:** Physicochemical properties of emu oil.

	Emu Oil	Lard	Tallow
Item	Concentration		
Acid value (mg/g KOH)	1.24 ± 0.49	0.63 ^a^	1.07 ^a^
Free fatty acid content (%)	0.80 ± 0.07	10.03 ^b^	3.19 ^b^
MDA (mg/100 g oil)	0.02 ± 0.01	0.05 ^c^	N/A
Peroxide value (meq/kg oil)	1.50 ± 1.46	3.67 ^d^	0.65 ^d^
Iodine value (g/100 g oil)	72.67 ± 2.08	77.90 ^a^	45.30 ^a^
Unsaponified matter (%)	0.54 ± 0.13	12 ^e^	0–0.5 ^e^
Refractive index (40 °C)	1.46 ± 0.01	1.45–1.46 ^e^	1.45–1.46 ^e^
Carotenoid content (mg/kg oil)	5.92 ± 0.62	N/A	N/A
Total phenolic content (mg GAE/kg oil)	6.64 ± 0.37	N/A	N/A

^a–e^: data source; ^a^ [25]; ^b^ [26]; ^c^ [24]; ^d^ [27]; ^e^ [28]. N/A: data not available.

**Table 2 nutrients-14-02238-t002:** Fatty acid composition of emu oil.

	Emu Oil	Lard ^a^	Tallow ^b^
Fatty Acid	% Total Fatty Acids		
Saturated fatty acid	34.78 ± 1.04	52.10	48.00
Palmitic acid (C16:0)	25.67 ± 1.11	26.86	28.40
Heptadecanoic acid (C18:0)	9.06 ± 0.04	N/A	14.80
Docosanoic acid (C22:0)	0.05 ± 0.03	0.27	14.8
Unsaturated fatty acid	64.28 ± 1.04	47.41	52.00
Palmitoleic acid (C16:1 cis)	4.08 ± 1.11	1.52	4.7
Heptadecenoic acid (C17:1)	0.06 ± 0.03	N/A	N/A
Oleic acid (C18:1 cis)	45.76 ± 0.53	33.71	44.6
Linoleic acid (C18:2 cis)	14.00 ± 0.21	10.90	2.7
Linolenic acid (C18:3 cis)	0.37 ± 0.28	0.49	N/A

^a^ and ^b^: data source; ^a^ [29]; ^b^ [30]. N/A: data not available.

**Table 3 nutrients-14-02238-t003:** Bacterial reverse mutation assay of emu oil.

Treatment (μg/Plate)	TA97a		TA98		TA100		TA102		TA1535	
	−S9	+S9	−S9	+S9	−S9	+S9	−S9	+S9	−S9	+S9
Revertants per plate ^a^										
Main test										
Vehicle control ^b^	183 ± 3	183 ± 9	17 ± 1	19 ± 7	167 ± 10	190 ± 20	358 ± 14	268 ± 44	17 ± 4	21 ± 7
5000	196 ± 13	184 ± 18	20 ± 13	18 ± 2	161 ± 22	185 ± 46	289 ± 18	243 ± 27	19 ± 3	16 ± 3
1581	186 ± 17	191 ± 20	20 ± 5	16 ± 5	175 ± 15	160 ± 13	296 ± 63	246 ± 14	16 ± 3	12 ± 3
500	177 ± 12	165 ± 14	17 ± 2	30 ± 18	167 ± 17	162 ± 13	317 ± 31	197 ± 10	19 ± 1	14 ± 1
158.1	183 ± 6	163 ± 16	23 ± 4	21 ± 7	155 ± 28	174 ± 19	278 ± 76	208 ± 20	17 ± 6	16 ± 3
50	159 ± 28	162 ± 26	15 ± 4	23 ± 5	161 ± 19	166 ± 27	276 ± 39	243 ± 22	17 ± 2	13 ± 2
Positive control ^c^	1556 ± 156 *	715 ± 211 *	947 ± 249 *	1365 ± 171 *	1145 ± 123 *	876 ± 27 *	3501 ± 649 *	1318 + 72 *	75 ± 12 *	258 ± 39 *
Confirmatory test										
Vehicle control ^b^	181 ± 10	163 ± 18	17 ± 3	15 ± 4	171 ± 29	149 ± 44	389 ± 51	359 ± 26	21 ± 6	14 ± 4
5000	176 ± 7	199 ± 4	13 ± 3	19 ± 4	171 ± 26	158 ± 29	413 ± 19	415 ± 32	13 ± 2	14 ± 2
1000	153 ± 5	173 ± 23	20 ± 5	12 ± 2	178 ± 17	151 ± 30	383 ± 22	375 ± 24	14 ± 4	16 ± 6
200	166 ± 17	173 ± 5	19 ± 3	16 ± 1	145 ± 24	175 ± 17	389 ± 32	302 ± 67	10 ± 6	14 ± 4
40	158 ± 9	176 ± 4	18 ± 5	15 ± 4	177 ± 12	157 ± 9	377 ± 44	282 ± 52	13 ± 5	16 ± 2
8	154 ± 14	170 ± 9	17 ± 4	18 ± 4	199 ± 21	142 ± 36	395 ± 17	317 ± 9	8 ± 3	14 ± 3
Positive control ^c^	1237 ± 223 *	565 ± 53 *	739 ± 119 *	1000 ± 66 *	796 ± 243 *	947 ± 104 *	1936 ± 354 *	1076 ± 76 *	91 ± 11 *	104 ± 14 *

^a^ All numbers represent means ± standard deviation (n = 3). ^b^ The vehicle control was 100 uL DMSO. ^c^ Positive controls for each strain were: for strains TA97a and TA100/−S9, 4-nitro-1,2-phenylenediamine monohydrochloride (40 μg/plate); TA98/−S9, 2-aminofluorene (5 μg/plate); for TA102/−S9, mitomycin C (1 μg/plate); for TA1535/−S9, 4-nitroquinoline-N-oxide (1 μg/plate); for strains TA97a, TA98, TA100 and TA1535/+S9, 2-aminofluorene (5 μg/plate); for TA102/+S9, 1,8-dihydroxyanthraquinone (50 μg/plate). * Significantly different from the vehicle control at *p* < 0.05.

**Table 4 nutrients-14-02238-t004:** Summary results of chromosome aberration assay in Chinese hamster lung (CHL) cells treated with emu oil.

Exposure Period (h)	Treatment	No. of Metaphases Scored		No. of Metaphases with Different Aberration Types			
			No. of Metaphases with Aberrations	Breaks	Exchanges	Gaps	
					Ring Chromosome	Chromatid Exchanges	
2 (Without S9 mix)	MEM (200 μL)	100	0	0	0	0	0
	Emu oil (1.25 μL/mL)	100	0	0	0	0	0
	Emu oil (2.50 μL/mL)	100	0	0	0	0	0
	Emu oil (5.00 μL/mL)	100	0	0	0	0	0
	Mitomycin C	100	10 **	1 **	3 **	25 **	0
2 (With S9 mix)	MEM (200 μL)	100	0	0	0	0	0
	Emu oil (1.25 μL/mL)	100	0	0	0	0	0
	Emu oil (2.50 μL/mL)	100	0	0	0	0	0
	Emu oil (5.00 μL/mL)	100	0	0	0	0	0
	Cyclophosphamide	100	10 **	0	1 **	12 **	0

Abbreviations: MEM, Minimum Essential Media. ** Significant difference from the MEM at *p* < 0.01.

**Table 5 nutrients-14-02238-t005:** Summary results of micronucleus test conducted on emu oil.

Treatment	Dose	No. of Mice	Rate of PCE ^a^ (Mean ± SD%)	Rate of MPCE ^b^ (Mean ± SD‰)
Male				
Vehicle control (corn oil)	0	5	58.2 ± 5.3	0.6 ± 0.2
Emu oil	5 mL/kg	5	51.6 ± 5.7	0.8 ± 0.4
Emu oil	10 mL/kg	5	58.7 ± 11.9	0.9 ± 0.7
Emu oil	20 mL/kg	5	54.7 ± 7.3	1.3 ± 0.8
Positive control (cyclophosphamide)	40 mg/kg	5	56.7 ± 6.2	22.5 ± 11.9 **
Female				
Vehicle control (corn oil)	0	5	49.5 ± 5.1	0.5 ± 0.4
Emu oil	5 mL/kg	5	56.0 ± 3.3	0.9 ± 0.7
Emu oil	10 mL/kg	5	56.9 ± 8.4	0.7 ± 0.4
Emu oil	20 mL/kg	5	53.1 ± 5.3	1.6 ± 1.2
Positive control (cyclophosphamide)	40 mg/kg	5	55.6 ± 4.9	21.0 ± 5.5 **

^a^ PCE = polychromatic erythrocytes. ^b^ MPCE = number of micronucleated polychromatic erythrocytes observed per 2000 polychromatic erythrocytes examined. ** Significant difference from the cyclophosphamide at *p* < 0.01.

**Table 6 nutrients-14-02238-t006:** General appearance and behavioral observations of the animal treated orally with emu oil in the acute test.

Observation	Male	Female
Eye color	No effect	No effect
Urination	Normal	Normal
Rate of respiration	Normal	Normal
Change in skin	No effect	No effect
Diarrhea	Not present	Not present
General physique	Normal	Normal
Death	Alive	Alive

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
