# Peer review of "Chemical Characterization and In Vivo Toxicological Safety Evaluation of Emu Oil"

_nutrients, 2022, doi:10.3390/nu14112238_

Round 1
Reviewer 1 Report
The study of Lan et al. examined the chemical characterization and in vivo toxicological safety evaluation of emu oil. The study is well-designed and all the experiments are thorough. However, many changes concerning text editing and English language are needed. Listed below there are some examples.
Page 1, line 32: "There is" instead of There are"
Page 1, line 35: “in” instead of “with”
Page 1, line 39: “States” instead of “State”
Page 1, line 39: “endows emu oil with” instead of “endows emu oil”
Page 2, line 48-49: Rephrase as “Therefore, emu oil has been shown to prevent and treat gastrointestinal inflammation and bone problems after oral administration to animal models”
Page 2, line 92: Please refer the kit used (company, origin)
Page 3, line 101, 110 and 118: delete “The”
Page 3, line 120: “at room temperature” instead of “at the room temperature”
Page 3, line 122: delete “And”
Page 10, line 325-326: Please rephrase
Page 10, line 332: “as shown in Table 1”
Page 10, line 333: “shown in Table 2” instead of “from Table 2”
Page 10, line 333: Rephrase the sentence “Lower saturated fatty acid content makes it a strong resistant to oxidative rancidity”
The paper needs extensive editing of English language.
Author Response
Please see the attachment, thank you.

Reviewer 2 Report
Review of manuscript 1650396 for “Nutrients” journal
The reviewer believes that this is a well-structured manuscript presenting some original research data about the chemical characterization and in vivo toxicological safety evaluation of emu oil. The authors have considered the most relevant and updated literature evidence in this scientific area and present their results and analysis in a clear and meaningful way. However, a good quality/editorial check is kindly recommended since language can improve in certain parts. A few minor comments for the consideration of authors are given below:
Introduction:
- Line-39/editorial: “Unites States” (add “s”)
- In the last paragraph, please add a couple of sentences to highlight the “innovative” aspect of the paper (added-value in this field?)…what has been examined for very first time or in depth by this manuscript (e.g. mutagenicity/toxicity of emu oil?) compared to state of art in this research area?
Material and methods:
-Sections [2.2]- Extraction of emu oil by super critical fluid extraction, [2.6.4]-Micronucleus test. Please include a reference for each method (unless developed in this laboratory/research team….if so, please note explicitly).
-Line 109: “Determination of total phenolic content was according to a previous method-15 ….”…Please check…there must be a mistake….do you mean that same method applied for determination of total carotenoid and phenolic contents? Does Ref 15 concern Folin-Ciocalteou method? Please clarify!
- Section [2.7]-Statistical analysis: Maybe to add a sentence such as “statistical differences/order of activity expressed via use of manuscript letters-for the different treatments (such as: a<b<c…?).
Discussion:
- Line 325-326: “ As a results, the peroxide value was usually to investigated the primary oxidation of emu oil…” For me the sentence lack clarity…perhaps to delete and note, instead, that “peroxide value provides a quality indicator of the oils/reflecting the status of sensory/organoleptic value” or something similar?
- Line 344: “Moreover, it has biological components such as polyphenols, carotenoid, etc.”…Can you please briefly note what are the main carotenoid and phenolic compounds expected to be present in emu oil….and if you did not apply any analysis, to quote from literature?
- Line 378-379: “In summary, results of the present study were decisive in adoption of emu oil as an 378 alternative source of edible oil..”….For me not the best way to conclude on the manuscript..Maybe to highlight in a short paragraph the benefit of the findings/novelty? Do they trigger any potential for market application? Any need for follow up work (optimisation of critical parameters?)
Author Response
Please see the attachment, thank you.
